# Skeletal Muscle Dysfunction in Experimental Pulmonary Hypertension

**DOI:** 10.3390/ijms231810912

**Published:** 2022-09-18

**Authors:** Kosmas Kosmas, Zoe Michael, Aimilia Eirini Papathanasiou, Fotios Spyropoulos, Elio Adib, Ravi Jasuja, Helen Christou

**Affiliations:** 1Department of Pediatric Newborn Medicine, Brigham and Women’s Hospital and Harvard Medical School, Boston, MA 02115, USA; 2Department of Pediatrics, Boston Children’s Hospital and Harvard Medical School, Boston, MA 02215, USA; 3Department of Medicine, Pulmonary Division, Brigham and Women’s Hospital and Harvard Medical School, Boston, MA 02215, USA; 4Department of Medicine, Brigham and Women’s Hospital and Harvard Medical School, Boston, MA 02215, USA

**Keywords:** PAH, skeletal muscle, type II fibers, FoxO1

## Abstract

Pulmonary arterial hypertension (PAH) is a serious, progressive, and often fatal disease that is in urgent need of improved therapies that treat it. One of the remaining therapeutic challenges is the increasingly recognized skeletal muscle dysfunction that interferes with exercise tolerance. Here we report that in the adult rat Sugen/hypoxia (SU/Hx) model of severe pulmonary hypertension (PH), there is highly significant, almost 50%, decrease in exercise endurance, and this is associated with a 25% increase in the abundance of type II muscle fiber markers, thick sarcomeric aggregates and an increase in the levels of FoxO1 in the soleus (a predominantly type I fiber muscle), with additional alterations in the transcriptomic profiles of the diaphragm (a mixed fiber muscle) and the extensor digitorum longus (a predominantly Type II fiber muscle). In addition, soleus atrophy may contribute to impaired exercise endurance. Studies in L6 rat myoblasts have showed that myotube differentiation is associated with increased FoxO1 levels and type II fiber markers, while the inhibition of FoxO1 leads to increased type I fiber markers. We conclude that the formation of aggregates and a FoxO1-mediated shift in the skeletal muscle fiber-type specification may underlie skeletal muscle dysfunction in an experimental study of PH.

## 1. Introduction

Pulmonary arterial hypertension (PAH), a progressively debilitating disease, seriously compromises the patient’s quality of life and it is ultimately fatal. The primary pathobiology of it involves an increased pulmonary vascular resistance and right ventricular hypertrophy (RVH) and thus, its failure. Accumulating evidence supports that intrinsic skeletal and respiratory muscle dysfunction also occurs and this often precedes a hemodynamic compromise, thereby, further limiting exercise tolerance. Skeletal muscle atrophy, a decreased proportion of slow oxidative (Type I) to fast anaerobic (glycolytic) (Type II) fibers, reduced capillarity, and decreased muscle oxidative capacities are associated with decreased endurance and were previously reported in a human and experimental study of, PH but the underlying molecular mechanisms of it are unknown [1,2,3,4,5,6]. Furthermore, the effects of current therapies on skeletal muscle dysfunction are not fully understood. 

Given that exercise intolerance is a negative PAH prognostic factor, the delineation of the molecular basis for skeletal muscle dysfunction will likely yield novel therapeutic targets to improve the functional outcomes. Muscle weakness and decreased endurance may be due to atrophy, impaired regeneration, and metabolic dysfunction. The transcription factor FoxO1, which is known to integrate inflammatory and metabolic signals [7], has diverse effects on many cellular compartments that are pertinent to PAH and skeletal muscle differentiation and fiber specification. A member of the FoxO forkhead type transcription factors, FoxO1, which is known to be involved in PAH pathogenesis, is also expressed in skeletal muscle, and has important roles in differentiation and skeletal fiber type specification. FoxO1 is involved in the regulation of skeletal muscle energy metabolism, protein breakdown, muscle regeneration and adaptation to exercise in other systemic diseases but has not been studied in PAH-associated skeletal muscle dysfunction [8,9,10]. 

In the well-established Sugen 5416/hypoxia rat model of PAH (SU/Hx-PH), a clinically relevant rodent model, we found a decreased endurance in association with increased skeletal muscle FoxO1 levels, an increased proportion of fast glycolytic (Type II) fibers to slow oxidative (Type I) fibers, and the formation of sarcomeric aggregates in some muscles. These may contribute to impaired exercise tolerance in an experimental PH study. 

## 2. Results

### 2.1. Reduced Exercise Endurance in Rats with SU/Hx-Induced PH 

As previously reported [11,12,13], the right ventricular systolic pressure (RVSP) was significantly elevated in SU/Hx-induced animals when compared to that of the normoxic controls (Figure 1A). There was no difference in the left ventricular systolic pressure (LVSP) between the two groups of animals (Figure 1B). The SU/Hx rats exhibited a right ventricular hypertrophy (RVH), which was assessed by Fulton’s Index (FI), when compared to those of the controls (Figure 1C). To evaluate the contribution of a hemodynamic compromise to exercise intolerance, we performed exercise endurance tests in the same group of animals. As shown in Figure 1D, the SU/Hx rats demonstrated decreased endurance, with significantly shorter times until they reached exhaustion, when compared to those of the controls.

### 2.2. Increased Levels of Type II Markers in Soleus of Animals with SU/Hx-Induced PH

To define the extent of the skeletal and diaphragmatic muscle dysfunction, we harvested the soleus (predominantly slow, Type I fibers), the extensor digitorum longus (predominantly fast, Type II fibers), and the diaphragm (mixed fibers) from the experimental and control animals. We evaluated the expression of the Type I and Type II fiber-specific genes, the markers of metabolic function, and the mitochondrial biogenesis. We found, overall, that there were decreased Type I markers (*Actn2* and *Tnni1*) and decreased markers of mitochondrial biogenesis (*Tfam*, *PGC-1**α* and *Sirt-1*) in both the diaphragm and the soleus in the SU/Hx animals when these were compared to those of the controls (Figure 2A,B). We also found that there was a significantly increased expression of the Type II markers (*Myh1* and *Myh4*) in the soleus of the SU/Hx animals (Figure 2B). As expected, in the predominantly glycolytic “fast” Type II EDL muscle, we found no significant differences in the Type I, Type II, or mitochondrial biogenesis markers between the SU/Hx and the control animals (Figure 2C). Immunofluorescence studies confirmed that there were high expression levels of the Type II markers (MYH1, MYH4) in the soleus of the SU/Hx animals when compared to those of the controls (Figure 2D). There was a trend of increased Type II markers protein levels in the soleus of the SU/Hx animals when compared with that of the control animals, as shown in the immunoblots in Figure 2E (antibody recognizes myosin type II muscle fibers (MYH1/2/4/6)).

### 2.3. Sarcomeric Aggregates in Skeletal Muscles and Hearts from Animals with SU/Hx-Induced PH

We next investigated the structural properties of the sarcomeres and found sarcomeric disarrays in the skeletal muscles of the SU/Hx animals. In the soleus muscles from the control animals, we observed well-formed regular sarcomeres, whereas, the soleus sections from the SU/Hx animals showed a mixed sarcomeric organization. Some areas had well-formed sarcomeres, and in other areas, there were sarcomeric aggregates which are shown in the longitudinal sections in Figure 3A. Typically, in the skeletal muscle, the desmin intermediate filaments are located underneath the sarcolemma [14]. Immunofluorescence using desmin and troponin I antibodies revealed atypical desmin aggregates and an assembly of troponin components along the sarcolemma in the transverse sections of the soleus muscles from the SU/Hx animals when these were compared to those of the controls (Figure 3B). We also performed immunofluorescence studies to investigate the sarcomeric organization in whole heart specimens from the SU/Hx and control animals. Immunofluorescence using desmin and troponin I antibodies revealed elongated and well-organized sarcomeres in the control animals. In the whole heart specimens from the SU/Hx animals, we also saw a striated staining pattern for desmin and troponin-I, however, it was frequently irregular, and in addition, we detected areas with deposition of desmin aggregates, as is observed in desminopathies [15]. Aggregate-like structures were also seen when we stained the samples for the detection of troponin-I (Appendix A). In addition, in the SU/Hx rats, the M-line was severely disturbed in many areas of the cardiac sections as was detected by myomesin 1 (Appendix A).

### 2.4. Muscle Atrophy in Rats with SU/Hx-Induced PH 

We next asked whether muscle wasting plays a role in the skeletal muscle dysfunction of the SU/Hx-induced rats with PH. We evaluated the markers of protein degradation and muscle atrophy, Murf-1 and Atrogin, respectively [4,16], by employing a q-PCR. Neither marker was increased in the soleus or the diaphragm of the SU/Hx animals, thus supporting a lack of muscle atrophy. In fact, the amount of *Murf-1* mRNA was significantly decreased in the soleus muscle of the SU/Hx animals when this was compared to that of the controls, and *Atrogin-1* mRNA levels were significantly decreased in the diaphragm of the SU/Hx animals when they was compared to those of the controls (Figure 4A).

We then evaluated the muscle fiber’s cross-sectional area and the number of fibers per field that there was in the soleus of all the members of both rat groups. We found no significant difference among the groups in either the fiber cross-sectional area or in the number of fibers per field (Figure 4B). In a separate cohort of animals, there was a significant reduction in the soleus weight when it was expressed relative to the length of the tibia in the SU/Hx animals and when it was compared to that of the controls (Figure 4C). Taken together, these data indicate the absence of muscle atrophy in the diaphragm of the SU/Hx animals (Figure 4A), while the soleus atrophy may contribute to impaired endurance in the SU/Hx animals, along with other mechanisms. 

### 2.5. Increased Levels of Type II Markers in L6 Myotubes

In order to gain further insight into the mechanism that leads to a muscle type fiber switch in PAH, we used the well-characterized L6 rat myoblast cell line. We differentiated the L6 myoblasts to myotubes with a serum starvation method, and confirmed the fusion macroscopically (Figure 5A), and these resulted in the decreased abundance of Glut1 and the increased abundance of Glut4 at the RNA and protein levels, respectively (Figure 5B,C), as was previously known [17]. We observed that under these experimental conditions, there was an increased abundance of the Type II marker MYH1/2/4/6 expression levels, but there was no difference in the Type I marker Hexokinase II (Figure 5C).

### 2.6. Increased Levels of FoxO1 in Soleus Muscles of Rats with SU/Hx-Induced PH, and FoxO1 Inhibition Increases Type I Markers in L6 Myotubes

Once we established the L6 model of the muscle fiber type differentiation, we then proceeded to investigate the role of FoxO1, a transcription factor that is known to be involved in PAH pathogenesis and the regulation of skeletal muscle differentiation. We first evaluated the skeletal muscle expression of *FoxO1* in the SU/Hx animals. As previously reported, in healthy animals, there is higher abundance of *FoxO1* in the fast muscles (EDL) when it is compared to that of the slow (soleus) muscles (Figure 6A). Despite there being a lower FoxO1 expression in the other tissues in an experimental PH study [18], we found that in the SU/Hx model *FoxO1* mRNA levels were not only preserved, but there was in fact a trend for increased *FoxO1* mRNA levels in the soleus (Figure 6B) and the diaphragm (Figure 6C) of the SU/Hx rats when they were compared to those of the controls. In the differentiated L6 cells with increased levels of *myogenin* (a late marker of myoblast fusion [19]), we found increased levels of *FoxO1* mRNA (Figure 6D). Of note, we found a higher abundance of nuclear FoxO1 (active form) in the myoblasts when they were compared to the myotubes (Figure 6E). We also observed the cytoplasmic localization of phospho-FoxO1 (inactive form) in both the myoblasts and the myotubes. We then treated the L6 myoblasts with a FoxO1 inhibitor (AS1842856) to determine whether FoxO1 mediates the fiber type specification during myotube formation in the differentiated L6 cells. We found that after 48 hrs of the AS1842856 treatment, there was an abundance of myotube formation, macroscopically, that was accompanied by the increased levels of the type I muscle fiber marker, Hexokinase II [20], and unchanged levels of the type II muscle fiber marker, MYH1/2/4/6 (Figure 6F). This finding supports that FoxO1 may promote a decreased Type I/Type II ratio by inhibiting the Type I fiber specification.

### 2.7. Transcriptional Signature of Skeletal Muscle Dysfunction in SU/Hx-Induced PH

To identify the additional molecular mechanisms that are underlying skeletal muscle dysfunction in an experimental PH study, we used RNA sequencing to characterize the transcriptomic profile of the soleus, the EDL, and the diaphragm of the SU/Hx animals, and evaluated the differential gene expression of this group as compared to that of the control animals. There were distinct transcriptomic signatures in the diaphragm of the SU/Hx animals when these were compared to those of the controls, with 319 upregulated and 217 downregulated genes, and in the EDL, there were 94 upregulated and 77 downregulated genes with a false discovery rate (FDR) of <5%. (Figure 7). In the soleus, the transcriptomic profile analysis showed that there were only 17 upregulated and two downregulated genes with a false discovery rate (FDR) of <5% (Figure 7). Interestingly, the two most highly upregulated genes in all the muscles that were studied were *Cyp1a1* and *Cyp1b1*, and this finding was validated by a quantitative PCR (qPCR) in the soleus, the EDL, and the diaphragm from a separate cohort of SU/Hx-induced and control animals (Table 1). We then performed a gene set enrichment analysis (GSEA) with a focus on the gene ontology pathways for the diaphragm and found that the most significantly downregulated transcripts were related to mitochondrial function and the mitochondrial membranes (Figure 8C,D). The transcripts that were related to the contractile apparatus were also significantly downregulated (Figure 8E), while the most significantly upregulated pathways were related to inflammation and collagen metabolism (Figure 8A,B; Appendix A). 

## 3. Discussion

Exercise intolerance is a key contributor to poor quality of life in patients with PAH [21]. Exertional fatigue and dyspnea in PAH are attributed to cardiorespiratory impairment, but intrinsic skeletal muscle dysfunction also occurs and often precedes a hemodynamic compromise, thereby, further limiting exercise tolerance PH [1,2,3,4,5,6]. Using the S/Hx model of PH, we report a shift from type I to type II markers in the soleus muscles in the animals with PH in an experimental study. We further demonstrate the formation of sarcomeric protein aggregates, increased levels of the transcription factor FoxO1, and some evidence of possible atrophy in the soleus muscles from the animals with SU/Hx-induced PH.

The soleus muscle is a slow-endurance muscle that is predominantly comprised type I fibers (88%) and 12% type II fibers in the rat [22]. Type I fibers (slow or endurance fibers) are predominantly oxidative, with abundant mitochondria, and Type II fibers (fast fibers) are predominantly glycolytic. We found an increased abundance of type II fibers in the soleus muscle from the SU/Hx model, which was accompanied by decreased exercise endurance. Our findings support that a switch from the slow-twitch oxidative muscle fibers to the fast-twitch glycolytic muscle fibers that are prone to fatigue may contribute to the skeletal muscle dysfunction and exercise intolerance in PH. The mechanism of this switch is unclear, but our results support that the transcription factor FoxO1 may play a role. Our results are in agreement with Enache et al. [2] and Moreira-Concalves [4] who found skeletal muscle mitochondrial dysfunction in rats with monocrotaline-induced PH [2], and also with Malenfant [3] who reported abnormal mitochondrial function and an increased type II/type I muscle fiber composition in patients with PAH [3,21]. Our findings are also in agreement with Batt [1] who reported both muscle atrophy and a decreased Type I/Type II ratio in skeletal muscles from patients with idiopathic PAH. This earlier report also found a selectively decreased cross-sectional fiber area in the Type I fibers and although our experimental design did not allow us to detect selective atrophy in Type I vs. Type II fibers, we did find evidence of soleus muscle atrophy that needs to be investigated further. A study by de Man et al. reported diaphragmatic atrophy in the MCT and SU/Hx models of PH, and impaired diaphragmatic contractile properties with a preserved oxidative function in the MCT model. In addition, these authors reported diaphragmatic atrophy and weakness in patients with PAH [5]. A milder phenotype of diaphragmatic dysfunction was also recently reported by Cannon et al. in the mouse model of SU/Hx-induced PH [23].

To our knowledge, sarcomeric dysfunction with desmin aggregates has not been previously described in an experimental PH study. In healthy skeletal muscles, desmin intermediate filaments are typically located at the level of the Z-disk and underneath the sarcolemma. We report that in the soleus muscles and hearts of the animals with the SU/Hx-induced PH, this typical pattern is lost and the desmin aggregates are apparent in the sarcoplasm and in the subsarcolemmal region. This pathologic pattern is seen in one mouse and two rat desminopathy-related models [14,24,25], and in patients with desmin mutations [26,27,28,29], and its significance in an experimental PH study needs to be further explored.

We used the L6 in vitro differentiation system for further mechanistic insights into muscle fiber specification. In this established system of myogenic differentiation, there is a shift in the glucose transporter isoform expression from predominantly Glut1 in myoblasts to predominantly Glut44, the insulin-regulatable glucose transporter isoform, in contracting myotubes [17]. In addition, Myogenin, another marker of differentiation is increased in the myotubes. We reproduced these findings as previously reported, and found that under our experimental conditions, there was increased abundance of the Type II markers Myh1,2,4. Given the important role and proposed therapeutic targeting of the transcription factor FoxO1 in PAH, we evaluated its potential role in the muscle fiber specification in the L6 system. FoxO1 is known to be in higher abundance in predominantly Type II (fast) muscles such as the EDL (also shown in Figure 6A). Consistent with this, we found that the in vitro differentiation of L6 cells was associated with a predominantly Type II fiber phenotype and an increased nuclear translocation of FoxO1. Furthermore, the inhibition of FoxO1 during L6 cell differentiation resulted in an increased Type I fiber marker expression which supports the fact that FoxO1 may mediate a shift in the muscle fiber type from Type I to Type II. The mechanisms that are underlying these observations need to be further defined, but they raise the important possibility that the therapeutic enhancement of FoxO1 in PAH may inadvertently contribute to skeletal muscle dysfunction through the direct effects of FoxO1 in skeletal muscle homeostasis. As previously reported, FoxO1 is decreased in multiple cellular compartments in PAH (including the lungs and vascular smooth muscle cells), and a therapeutic enhancement has been proposed to alleviate the cardiopulmonary compromise. The therapeutic enhancement of FoxO1 activity was ameliorated in an experimental PH study [18], but some of the agents that used were associated with skeletal muscle atrophy [16]. Our data support that FoxO1 is not decreased in the soleus muscle in the experimental PH study, in fact, we found a trend for increased FoxO1, and this was associated with a decreased abundance of Type I fibers. We therefore speculate that the systemic enhancement of the FoxO1 function may exacerbate skeletal muscle dysfunction by decreasing the proportion of Type I to Type II fibers. The potential detrimental role of FoxO1 activation in skeletal muscle function is supported by other studies in which skeletal muscle atrophy in MCT-induced PH is accompanied by increased FoxO1 levels [30], and FoxO1 inhibition attenuates the mechanical ventilation-induced diaphragmatic dysfunction in an experimental study of rats by inhibiting diaphragmatic atrophy [31].

Numerous studies have reported multiple and often conflicting functions of FoxO1 on muscle differentiation. The skeletal muscle overexpression of FoxO1 in transgenic mice leads to a reduced muscle mass, a reduced expression of slow fiber genes, and impaired muscle function [32], while the conditional ablation of FoxO1 in skeletal muscles results in a decreased proportion of slow fibers [8]. FoxO1 activity inhibited early myoblast differentiation in mouse C2C12 myoblasts [33], but this was required for myoblast fusion in rat L6 myoblasts [34]. Our discovery, that the inhibition of FoxO1 in L6 myotubes leads to an increased type I fiber abundance, adds to our understanding of the potential mechanisms that are contributing to muscle function impairment which are connected to FoxO1 activation.

To our knowledge, this is the first report of an RNAseq analysis in the skeletal muscles in an experimental PH study. We found a number of differentially expressed transcripts in the diaphragm and soleus muscles of the SU/Hx-treated animals when these were compared to those of the controls. The two most highly upregulated genes in the SU/Hx-PH group in all of the muscles that were studied were *Cyp1a1* and *Cyp1b1*, and this finding may be related to Aryl Hydrocarbon Receptor activation that was caused by Sugen the treatment, as was previously reported in the lung tissues and human pulmonary artery smooth muscle cells [35]. The significance of increased *Cyp1a1* and *Cyp1b1* when it is pertaining to skeletal muscle dysfunction in the experimental PH study needs to be further studied. Using gene ontology pathways, we found the significant upregulation of pathways that are related to inflammation and collagen metabolism, and the significant downregulation of the transcripts that are related to mitochondrial function and the contractile apparatus in the diaphragms of the SU/Hx animals. These findings need to be further explored, but they suggest that inflammatory and fibrotic pathways may contribute to impaired contractile properties in the skeletal muscles in the experimental study that was conducted on PH.

Taken together, our data support that decreased endurance in experimental pulmonary hypertension in the SU/Hx model is associated with diaphragmatic and skeletal muscle dysfunction through various mechanisms, including a shift in the skeletal muscle fiber type and muscle atrophy that may be mediated by FoxO1 and inflammatory pathways. This information needs to be taken into account when designing novel therapeutic approaches in order to optimize functional outcomes in PAH.

## 4. Materials and Methods

### 4.1. Animals and Experimental Models of Pulmonary Hypertension

All experimental procedures were conducted in accordance with the guidelines of the American Physiologic Society and the National Institutes of Health and were approved by the Harvard Institutional Animal Care and Use Committee and the Harvard Medical Area Standard Committee on Animals (Brigham and Women’s Institutional Animal Care and Use Committee). Adult (12-week-old) male Sprague–Dawley rats (250–300 g) were purchased from Charles River Laboratories (Wilmington, MA, USA) and acclimatized for 2–3 days prior to the experiments. PH was induced as previously described [11], with a subcutaneous injection of 20 mg/kg Sugen 5416 (Sigma, St. Louis, MO, USA) in dimethyl sulfoxide (DMSO; Sigma, St. Louis, MO, USA), then the rats were placed in a state of hypoxia (9% O_2_) for 3 weeks and returned to a state of normoxia. Oxygen was controlled to 9% ± 0.2% using an OxyCycler controller (BioSpherix, Redfield, NY, USA), and ventilation was adjusted using a fan and port holes to remove CO_2_ and ammonia. Control rats were injected with an equal volume of vehicle (DMSO) in a state of normoxia. The endpoints of the study were 24 days after injection. Hemodynamic measurements in anesthetized rats were performed.

### 4.2. Exercise Endurance Test

Animals were accustomed to exercise testing, and exercise endurance (time to exhaustion) was tested as previously reported [36,37]. Briefly, we used a standard three-lane rodent treadmill (Colombus Instruments, Columbus, OH, USA), and animals were accustomed to exercise testing by running on a rodent treadmill at slow speed (8–12 m/min) with no incline for one minute, three times a week, for 2 weeks. Exercise endurance was then tested on the same treadmill at 10 degrees incline and a speed of 15 m/min, as has been previously reported. The animals received a gentle electric stimulus if they stop running and when this occurred 3 consecutive times, this is defined as exhaustion. The time to exhaustion was a measure of exercise endurance and it was compared among the experimental groups. All testing was stopped at 90 min since the maximal running time was previously reported to be 90 min.

### 4.3. Hemodynamic Measurements

Hemodynamic measurements were performed as previously described [38]. Animals were anesthetized with an inhalation of 3% isoflurane, intubated through a tracheotomy, and mechanically ventilated on a rodent ventilator (Harvard Apparatus, tidal volume 1 mL/100 g body weight, 60 breaths per minute). The thoracic cavity was opened by incision of the diaphragm. A 23-gauge butterfly needle with tubing attached to a pressure transducer was inserted first into the right ventricle and then into the left ventricle, and pressure measurements were recorded using PowerLab monitoring hardware and software (ADInstruments, Colorado Springs, CO, USA). Mean RV systolic pressure (RVSP) and LV systolic pressure (LVSP) (in mmHg) over the first 10 stable heartbeats were recorded.

### 4.4. Rat Tissue Isolation

Post-hemodynamic measurements were taken when they were under deep anesthesia, and rats were sacrificed by exsanguination and tissues and organs were harvested immediately. Samples were snap-frozen and stored at −80 °C until processing.

### 4.5. Immunohistochemistry

Soleus muscles were frozen in an optimal cutting temperature compound (OCT, Sakura, Torrance, CA, USA) (cryosections 12 μm thick). Soleus samples and whole heart samples were also fixed in 4% paraformaldehyde before paraffin embedding was conducted. For immunofluorescence (paraffin sections, 5 μm thick), the samples were deparaffinized in 2 changes of xylene and rehydrated through a graded ethanol series, followed by antigen retrieval and antibody incubation. Sections were mounted, followed by the performance of confocal microscopy.

### 4.6. Cell Lines and Culture Conditions

L6 rat myoblast cells (Kerafast, Boston, MA, USA) were cultured in DMEM that was supplemented with 10% FBS, 100 mg/mL penicillin, and 100 mg/mL streptomycin. Cells were plated until reaching 80% confluence, after that, they were cultured for 7 days in either fresh DMEM with 10% FBS fetal bovine serum or DMEM containing 2% FBS to stimulate myotube formation. Cell lines were checked negative for Mycoplasma using the MycoAlert Mycoplasma Detection Kit (Lonza, Basel, Switzerland), according to the manufacturer’s instructions.

### 4.7. Isolation of mRNA and Transcript Expression Analysis

L6 cells were lysed, and diaphragm, EDL, and soleus samples were homogenized in TRIzol reagent (Invitrogen) per the manufacturer’s instructions for RNA isolation. RNA quantity and quality were determined using a NanoDrop 2000c spectrophotometer (NanoDrop Technologies, Wilmington, DE, USA). For cDNA synthesis, samples were first treated with DNase I (Sigma) for genomic DNA digestion and then the mRNA was reverse transcribed using the Superscript III (Invitrogen Life Technologies, Waltham, MA, USA) reagent. Real time quantification polymerase chain reaction (RT-qPCR) was performed using a StepOnePlus thermal cycler (Applied Biosystems, Waltham, MA, USA) using either the iTaq Universal SYBR Green Supermix (Bio-Rad, Hercules, CA, USA) or the PowerUp™ SYBR^®^ Green Master Mix kit. Primers were designed using nBLAST (National Center for Biotechnology Information, Bethesda, MD, USA) and PrimerQuest Tool (Integrated DNA Technologies, Coralville, IA, USA). The comparative Ct method (ΔΔCT) was used to calculate the relative gene expression level [39]. *Nucleoporin 133* (*Nup133*) and *GAPDH* were used as housekeeping genes for normalization. Primer sequences that were used in this study are presented in Appendix A.

### 4.8. RNA Sequencing and Bioinformatics Analysis

Soleus, EDL, and diaphragm muscles were dissected and snap frozen in liquid nitrogen. RNA extraction and quantification and sample integrity were performed by Novogene using Nanodrop spectrophotometer and Agilent 2100 bioanalyzer system, respectively. Library preparation and sequencing were performed by Novogene (Sacramento, CA, USA) using an Illumina Platform NovaSeq 6000 PE150. Reads (FASTQ files) that were obtained from paired-end sequencing were aligned to the Rattus Norvegicus 6.0 ENSEMBL genome using the standard parameters in the STAR version 2.5.4a [40]. Quality control was performed on raw reads and alignment results using FastQC as part of the VIPER workflow [41]. Sample clustering was visualized using principal component analysis. Cufflinks was used to assemble transcript-level expression data from filtered alignments [42]. Differential gene expression analysis was performed using DESeq2 version 1.26.0 using standard parameters, thus comparing SU/Hx animals with the controls. Counts were normalized using a negative binomial distribution as is standard with DESeq2. Experimental and control groups were compared using a Wald test in a pairwise comparison. Only the genes with a multiple-comparison that was adjusted to *p* < 0.05 were considered significant in the comparison. Significantly, differentially expressed genes were visualized using heatmaps and volcano plots and were used for downstream functional/pathway analyses using R bioconductor packages pheatmap and Enhanced Volcano. Pathway analysis was performed using Gene Set Enrichment Analysis (GSEA). Gene ontology pathway sets were tested for enrichment. The list of genes that were showing a significant upregulation or downregulation in the soleus, EDL, and diaphragm of SU/Hx animals were compared with controls and these comparisons were used to create Venn diagrams (http://bioinformatics.psb.ugent.be/webtools/Venn/, accessed on 10 February 2022).

### 4.9. Cellular Fractionation

L6 cells were washed and incubated with cytosolic buffer (20 mM HEPES (pH 7.8), 10 mM KCL, 1 mM MgCl_2_, 0.1% Triton, and 20% glycerol) for 10 min. Cells were scraped and spined at 2300 g for 5 min. Supernatants were cytosolic fractions. Pellets were resuspended in nuclear buffer (20 mM HEPES (pH 7.8), 1 mM EDTA, 20% glycerol, and 0.1% Triton, 400 mM NaCl) and incubated for 30 min. Samples were spined for at 14,000 g for 5 min. Supernatants were nuclear fractions. Both cytoplasmic and nuclear isolation buffers were supplemented with protease and phosphatase inhibitors. Equal fractions of lysates for both nuclear and cytoplasmic fractions were subjected to immunoblotting.

### 4.10. Western Blot Analysis

Cells and tissues were lysed using 1× RIPA buffer (Cell Signaling Technology, Danvers, MA, USA) containing protease and phosphatase inhibitors. Equal amounts of total protein were separated by electrophoresis on a 4–12% Bis-Tris gel and transferred to a polyvinylidene difluoride membrane. Blots were blocked in Tris Buffered Saline using Tween-20 (TBST) with 5% *w*/*v* nonfat dry milk, then incubated with primary antibodies overnight at 4 °C. The membranes were washed and then incubated with horseradish peroxidase–conjugated secondary antibodies at room temperature for 1 h. Bands were detected using SuperSignal West Pico Plus chemiluminescence substrate (Thermo Fisher Scientific, Waltham, MA, USA), photographed using Biorad ChemiDoc Imaging System and quantified using ImageJ analysis software.

### 4.11. Antibodies and Reagents

The following antibodies were used: skeletal muscle myosin (Santa Cruz Biotechnology, Dallas, TE, USA), Glut4 (Proteintech, Rosemont, IL, USA), Myomesin1 (Proteintech), Hexokinase II (Cell Signaling Technology), FoxO1 (Cell Signaling Technology), phospho-FoxO1 (Cell Signaling Technology) (rabbit), Histone H3 (Cell Signaling Technology), Glut1 (Cell Signaling Technology), GAPDH (Cell Signaling Technology), Vinculin (Cell Signaling Technology), Desmin (Milipore/Sigma-Aldrich), Troponin-I (Milipore/Sigma-Aldrich), b-Actin (mouse; Milipore/Sigma-Aldrich), anti-mouse HRP (GE Healthcare, Chicago, IL, USA), and anti-rabbit HRP (GE Healthcare). AS1842856 was purchased from Adooq.

### 4.12. Confocal Microscopy

Tissues were processed for immunofluorescence analysis and then incubated with primary antibodies overnight at 4 °C, washed, and incubated with fluorophore-conjugated secondary antibodies for one hour. Nuclei were visualized with DAPI (4′,6-diamidino-2-phenylindole) staining. Images were captured using a FluoView FV-10i Olympus Laser Point Scanning Confocal Microscope using a 60-objective lens. Confocal filters (Excitation/Emission nm) that were used for microscopy imaging were: 358/461 (DAPI), 490/525 (Alexa-Fluor488), 590/617 (Alexa-Fluor594), and 578/ 603 (Alexa-FluorA568). Antibodies were purchased from Invitrogen.

### 4.13. Statistical ANALYSIS

Statistical analyses were performed using GraphPad Prism version 8.4.3 (GraphPad Software, La Jolla, CA, USA). One-way ANOVA with Tukey’s post-test was used when comparing multiple groups, or Student’s t-test was used when comparing two groups. Data are presented as mean and standard error of the mean (SEM). Differences were considered statistically significant if *p* < 0.05.

## 5. Conclusions

The novel concept that is put forward here is that the decreased endurance that is observed in the experimental PH study correlates with the formation of sarcomeric aggregates in some muscles and increased FoxO1 levels that may drive a switch from slow oxidative (Type I) fibers to fast glycolytic (Type II) fibers, as depicted in the graphical abstract.

## Figures and Tables

**Figure 1 ijms-23-10912-f001:**
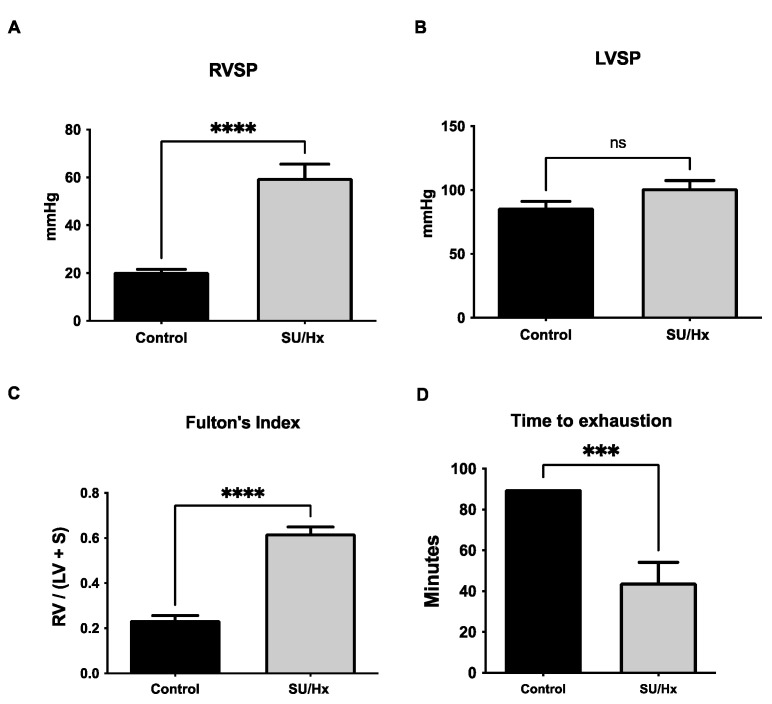
(**A**) Increased ventricular systolic pressure (RVSP) in SU/Hx-induced PH rats is seen, and (**B**) no significant difference on left ventricular systolic pressure (LVSP) can be seen when they are compared to those of the controls. (**C**) Right ventricular hypertrophy was assessed by Fulton’s Index (ratio of right ventricular weight to left ventricular and septal weight) in SU/Hx-induced PH rats and this is compared to that of the controls. (**D**) Decreased endurance for SU/Hx animals in treadmill exercise testing. Average time running for SU/Hx-induced PH rats is compared to that of the controls. Values are expressed as the mean ± SEM for six animals per group. Statistical analysis is conducted by a Student’s *t*-test, *** *p* < 0.001, **** *p* < 0.0001, ns: not significant.

**Figure 2 ijms-23-10912-f002:**
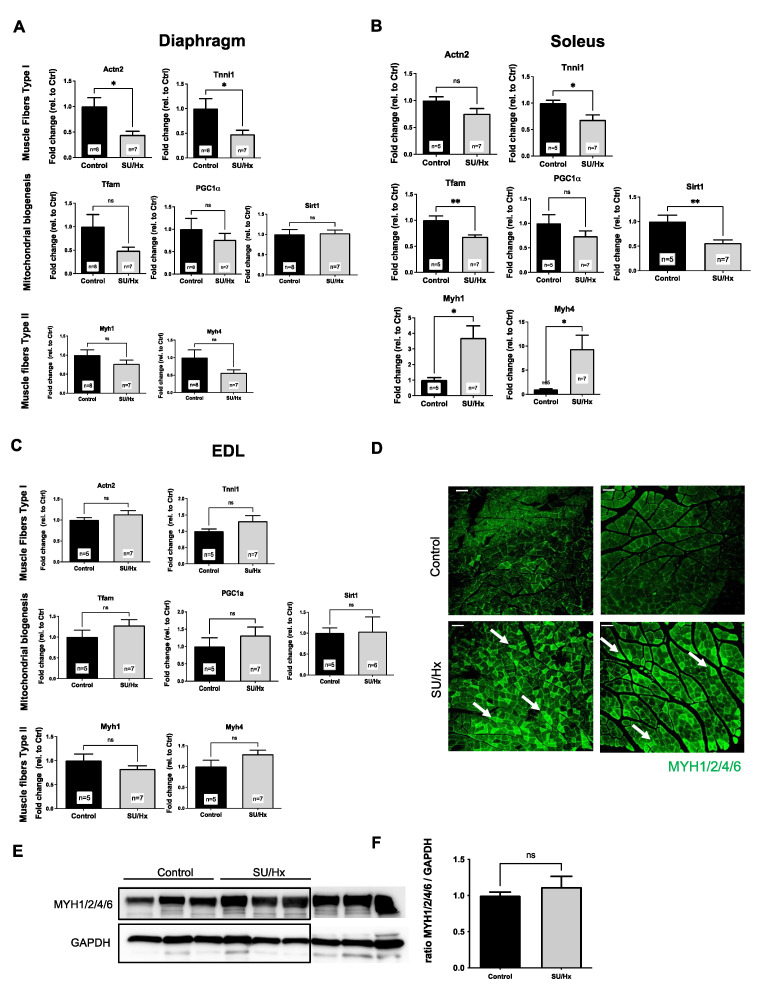
(**A**) Decreased abundance of Type I markers (*Actn2* and *Tnni1*) and markers of mitochondrial biogenesis (*Tfam* and *PGC-1α*) in diaphragms from animals with SU/Hx-induced PH when compared to those of the controls. (**B**) Decreased abundance of Type I markers (*Actn2* and *Tnni1*) and markers of mitochondrial biogenesis (*Tfam*, *PGC-1α*, *SIRT-1*) and increased abundance of Type II markers (*Myh1* and *Myh4*) in soleus muscles from animals with SU/Hx-induced PH when compared to those of the controls. (**C**) No difference in abundance of Type I markers (*Actn2* and *Tnni1*), markers of mitochondrial biogenesis (*Tfam, PGC-1α, SIRT-1*), and Type II markers (*Myh1* and *Myh4*) in EDL muscles from animals with SU/Hx-induced PH when compared to those of the controls. Expression is normalized to *Nup133*. Values are expressed as the mean ± SEM. Statistical analysis is conducted by a Student’s *t*-test, * *p* < 0.05, ** *p* < 0.01, ns: not significant. (**D**) In the soleus muscle from the control animals, there is less than 1% type II muscle fibers as compared to the 25% proportion that is seen when it was stained with Myh1/2/4/6 (arrows) in the soleus muscle from animals with SU/Hx-induced PH. Immunofluorescence microscopy of soleus cross sections from experimental animals (*n* = 3 animals per group). Scale bar, 100 μm. (**E**) Abundance of the Type II markers MYH1/2/4/6 in soleus from SU/Hx animals when compared to that of the controls. Immunoblot with GAPDH are used as a loading control and (**F**) quantitative analysis of MYH1/2/4/6 protein (*n* = 9 animals per group). Values are expressed as the mean ± SEM. Statistical analysis is conducted by a Student’s t-test, ns: not significant.

**Figure 3 ijms-23-10912-f003:**
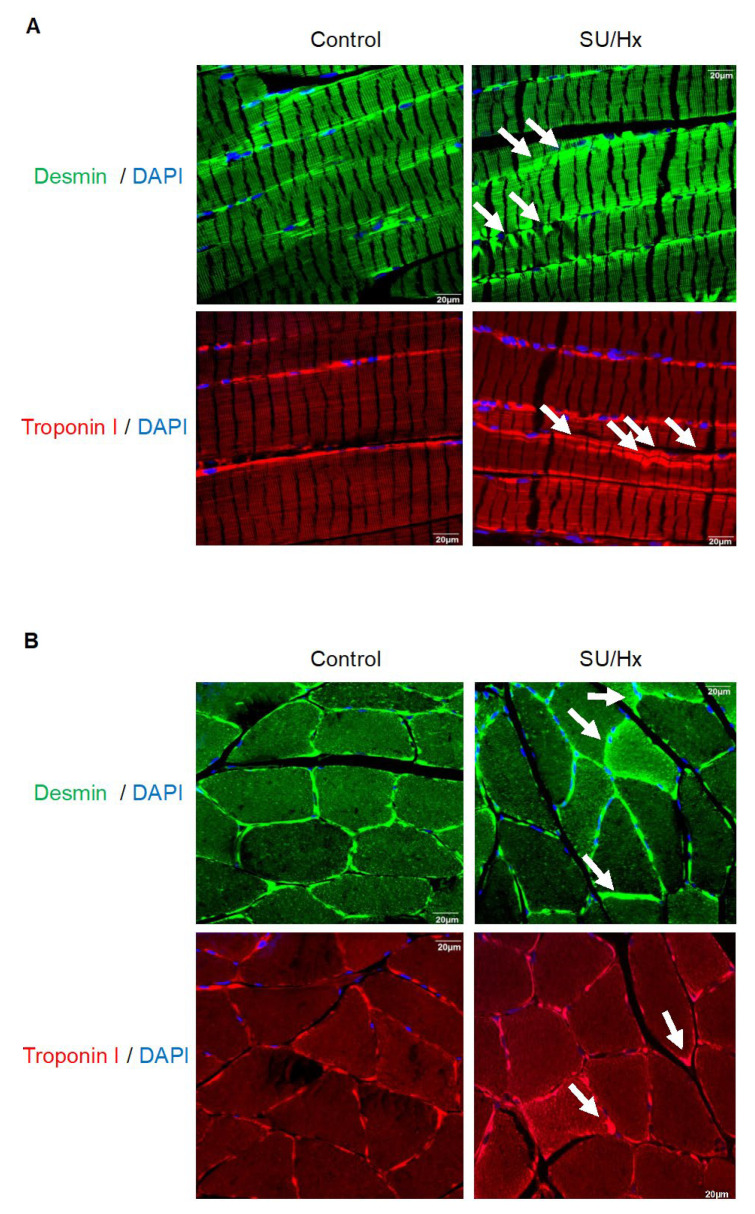
(**A**) Paraffin-embedded longitudinal sections of soleus muscle that is stained with desmin (Z-disc) and troponin-I specific antibodies showing a compromised and disorganized sarcomeric organization with aggregate-like structures (arrows) in SU/Hx-induced PH rats when they are compared to those of the controls. (**B**) Thick transverse desmin aggregates along the sarcolemma (arrows) and an assembly of troponin components along the sarcolemma (arrows) in transverse cryosections of soleus from animals with SU/Hx-induced PH is seen, and these are compared to those of the controls. Immunofluorescence microscopy with desmin and troponin-I specific antibodies, respectively. Nuclei are visualized after DAPI staining. Scale bar, 20 μm. *n* = 3 animals per group.

**Figure 4 ijms-23-10912-f004:**
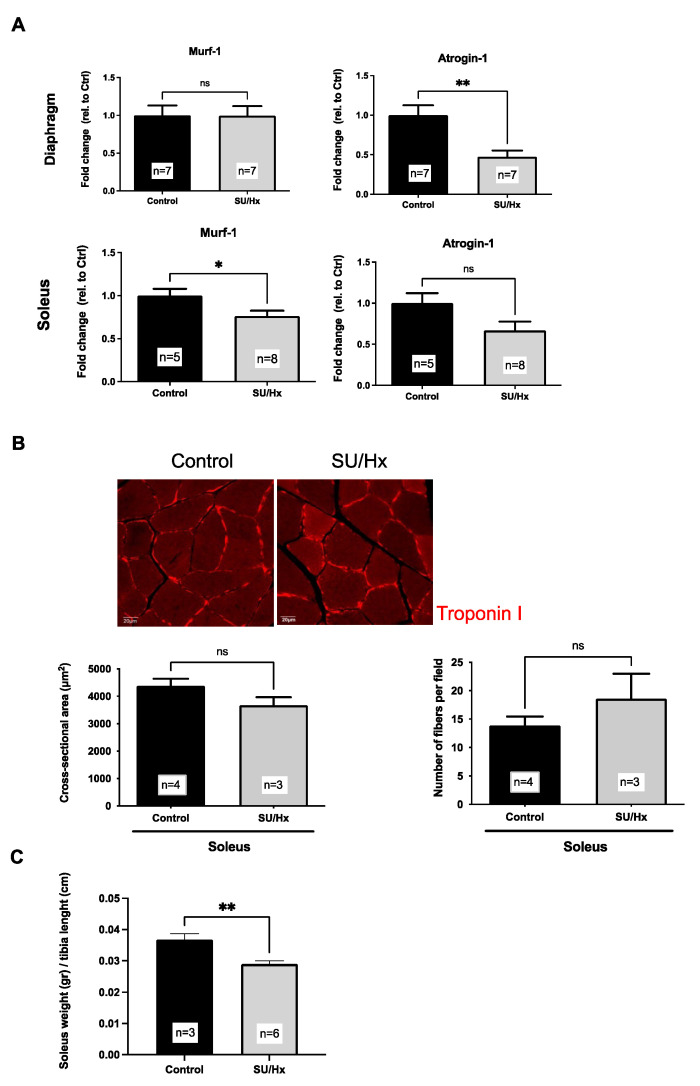
(**A**) No difference is seen in atrophy markers (*MuRf1*, *Atrogin*) in diaphragm and soleus muscle from SU/Hx-induced PH rats when they are compared to the controls. Expression normalized to *Nup 133*. (**B**) No difference is seen in average cross-sectional area and in the number of soleus muscle fibers per field for SU/Hx-induced PH rats when these are compared to those of the controls. Representative images of troponin I-stained soleus sections from SU/Hx-induced PH rats and controls. Scale bar, 20 μm. (**C**) Decreased ratio is seen in the soleus weight-to-tibia length from SU/Hx-induced PH rats when compared to that of the controls. Values are expressed as the mean ± SEM. Statistical analysis is conducted by a Student’s *t*-test, * *p* < 0.05, ** *p* < 0.01, ns: not significant.

**Figure 5 ijms-23-10912-f005:**
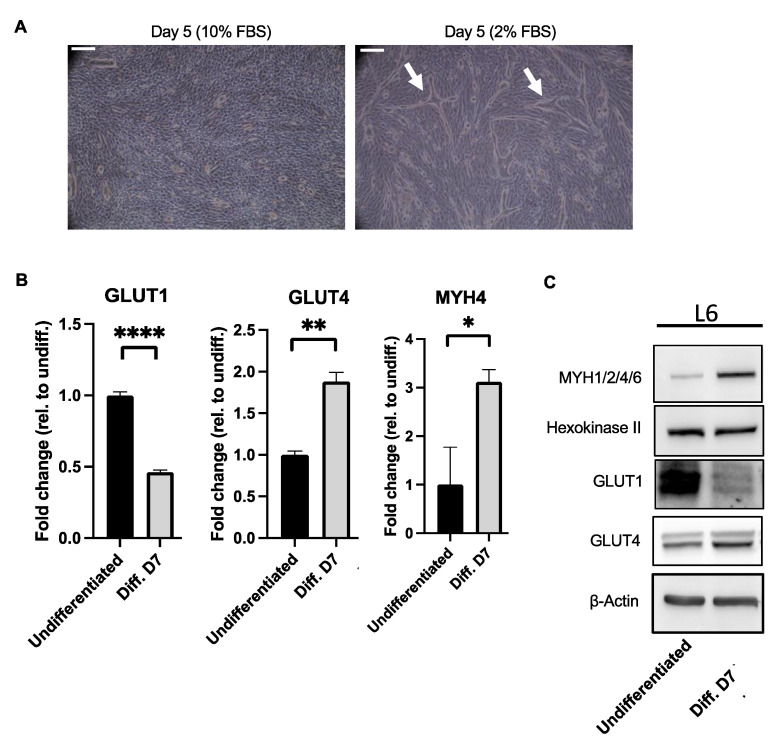
(**A**) L6 cells are differentiated in 2% fetal bovine serum media for 5 days. Myotubes structures are indicated with arrows. (**B**) Decreased RNA levels of *Glut1* and increased RNA levels of *Glut4* and *Myh4* in differentiated L6 cells (Day 7) are found when these are compared to the undifferentiated cells. Expression is normalized to *GAPDH*. Values are expressed as the mean ± SEM. Statistical analysis is conducted by a Student’s *t*-test, * *p* < 0.05, ** *p* < 0.01, **** *p* < 0.0001. (**C**) Immunoblot results are showing increased levels of Type II marker MYH1/2/4/6 and Glut4, decreased levels of Glut1 and unchanged levels of type I marker Hexokinase II in L6 lysates from differentiated L6 cells (Day 7) when these are compared to those of the untreated undifferentiated cells. Beta-actin was used as a loading control.

**Figure 6 ijms-23-10912-f006:**
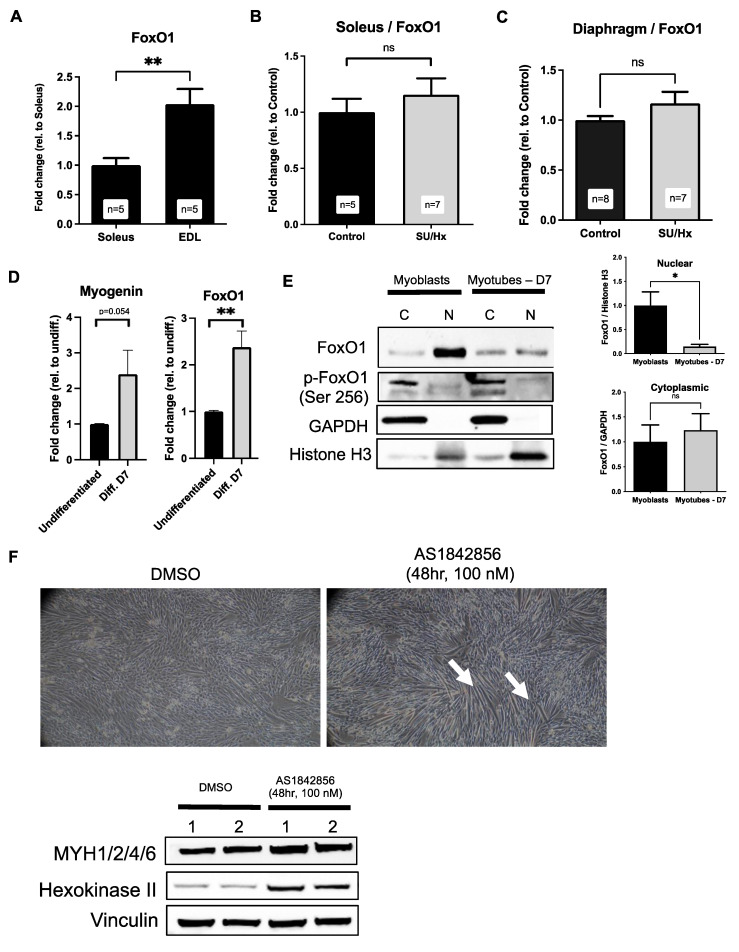
(**A**) There is an increased expression of *FoxO1* mRNA in the EDL (fast twitch) when this is compared to the soleus (slow twitch) from the control animals. (**B**) A trend for the upregulation of *FoxO1* mRNA in soleus of SU/Hx-induced PH is compared to that of the control animals. (**C**) A trend for the upregulation of *FoxO1* mRNA in diaphragm of SU/Hx-induced PH is compared to that of the control animals. Expression is normalized to *Nup 133*. Values are expressed as the mean ± SEM. Statistical analysis is conducted by a Student’s *t*-test, ** *p* < 0.01, ns: not significant. (**D**) There is an increased expression of *myogenin* and *FoxO1* mRNA in the whole cell lysates from differentiated L6 cells (Day 7) when this is compared to the whole cell lysates from undifferentiated cells. Expression is normalized to *GAPDH*. Values are expressed as the mean ± SEM. Statistical analysis was conducted by a Student’s *t*-test, ** *p* < 0.01. (**E**) Immunoblots and quantitative analysis showing that there was a cellular fractionation of the L6 myoblasts and myotube lysates. There was predominantly nuclear FoxO1 localization in myoblasts. There was cytosolic p-FoxO1 localization in myoblasts and myotubes. GAPDH was used as a marker for cytosolic fraction, and H3 was used as a marker for nuclear fraction. Values are expressed as the mean ± SEM. Statistical analysis was conducted by a Student’s *t*-test, * *p* < 0.05. (**F**) L6 myotube structure formation (indicated with arrows) after FoxO1 inhibitor treatment (AS1842856, 100 nM, 24 h). Immunoblots showing that there were unchanged levels of type II marker MYH1/2/4/6 and increased levels of type I marker Hexokinase in L6 myotube lysates after FoxO1 inhibitor treatment.

**Figure 7 ijms-23-10912-f007:**
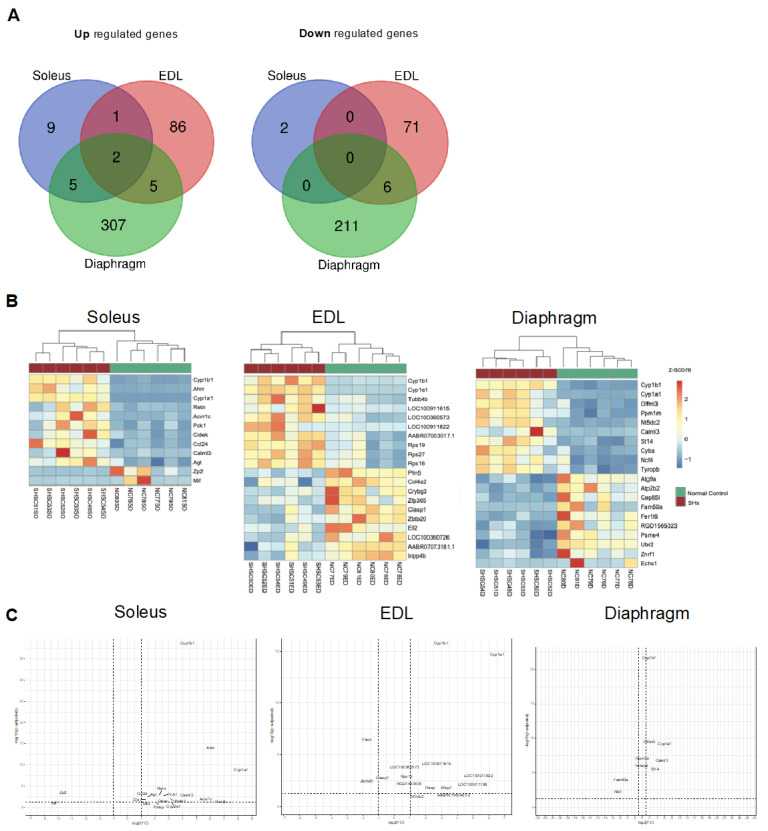
(**A**) Venn diagram showing the number of differentially expressed genes (DEGs) in the SU/Hx vs. control groups in three tissues: soleus, EDL, and diaphragm. (**B**) Hierarchical clustering analysis of the top 10 most significantly upregulated genes for soleus, EDL, and diaphragm, the two most significantly downregulated differential expressed genes that were identified in soleus, and the 10 most downregulated differential expressed genes in EDL and diaphragm between SU/Hx and control groups. (**C**) Volcano plot of the mRNA transcripts of soleus, EDL, and diaphragm in the SU/Hx vs. control groups. Significantly downregulated genes are in red, significantly upregulated genes are in green, nonsignificant genes are in gray. *n* = 6 individual animals per group.

**Figure 8 ijms-23-10912-f008:**
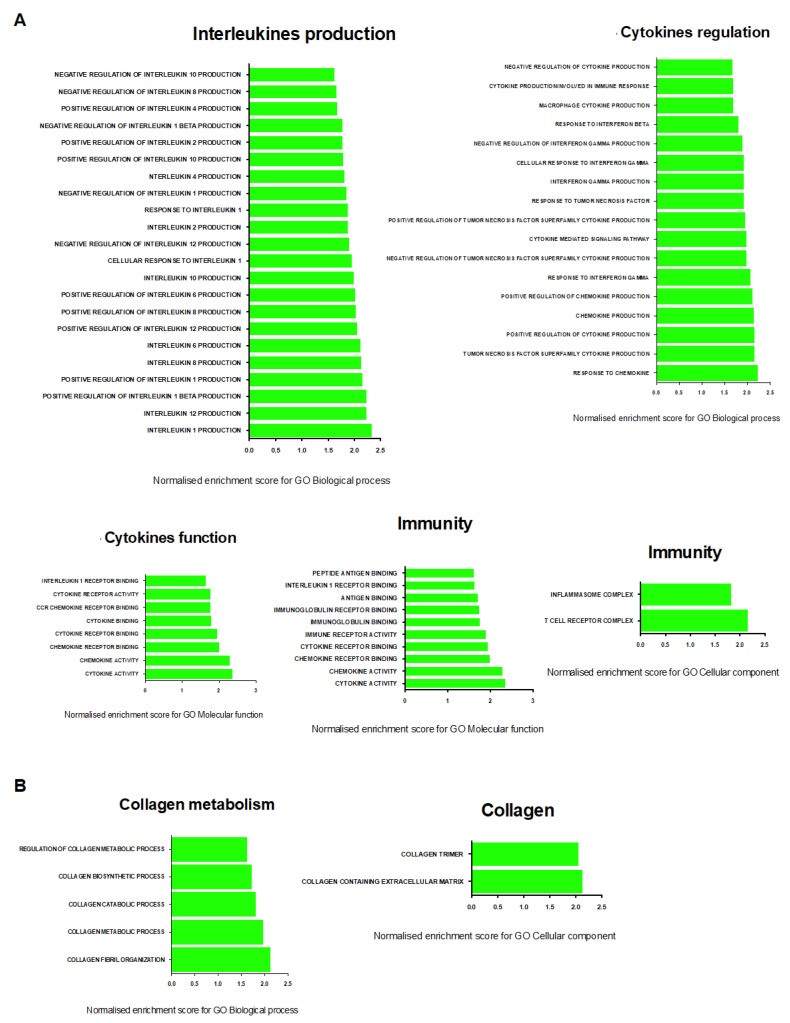
Gene ontology enrichment analysis of biological processes, cellular components, and molecular functions in diaphragm transcripts between SU/Hx and control groups. (**A**) The most significantly upregulated biological processes, interleukin production-related terms, cytokine regulation-related terms, and the most significantly upregulated molecular functions, cytokine-related terms, immunity-related terms, and the most significantly upregulated cellular components and immunity-related terms; (**B**) the most significantly upregulated biological processes, collagen metabolism-related terms, and the most significantly upregulated cellular components and collagen-related terms; (**C**) the most significantly downregulated biological processes and mitochondrial function-related terms; (**D**) the most significantly downregulated cellular components and mitochondrial membrane-related terms; (**E**) the most significantly downregulated biological processes, contractile apparatus-related terms, and the most significantly downregulated cellular components and contractile apparatus-related terms by false discovery rate (FDR).

**Table 1 ijms-23-10912-t001:** Validation of RNA sequencing. Top differentially expressed genes in the soleus, EDL, and diaphragm of SU/Hx vs. control rats are validated by qPCR in an independent cohort of animals (*n* = 5 per group).

	RNA Sequencing		PCR Validation	
Soleus				
**Gene**	**Fold Change**	** *p* ** **-Value adj.**	**Mean Fold Change**	** *p* ** **-Value**
*Cyp1a1*	371.05	2.49 × 10^−10^	11.56	0.0635
*Cyp1b1*	16.68	1.28 × 10^−38^	11.01	0.0151
**EDL**				
**Gene**	**Fold Change**	** *p* ** **-Value adj.**	**Mean Fold Change**	** *p* ** **-Value**
*Cyp1a1*	111.05	9.47 × 10^−16^	128.4	0.0155
*Cyb1b1*	6.54	3.43 × 10^−16^	0.8	0.7593
**Diaphragm**				
**Gene**	**Fold Change**	** *p* ** **-Value adj.**	**Mean Fold Change**	** *p* ** **-Value**
*Cyp1a1*	88.95	2.19 × 10^−10^	123.2	0.014
*Cyp1b1*	7.76	8.02 × 10^−23^	0.53	0.4874

## Data Availability

The data presented in this study are available in the article or Appendix A.

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
