# Peer review of "Skeletal Muscle Dysfunction in Experimental Pulmonary Hypertension"

_ijms, 2022, doi:10.3390/ijms231810912_

Round 1

Reviewer 1 Report

This is a comprehensive assessment of skeletal muscle dysfunction and metabolic, gene expression and histological changes in Sprague Dawley rats that develop pulmonary arterial hypertension when injured with Sugen/hypoxia administration, a well characterized model for PAH. The rats develop pulmonary hypertension, lose exercise capacity   They find that skeletal muscle, particularly those higher in Type I oxidative fibers become weaker, show histological changes of injury and intracellular rearrangement of contractile elements, and and show changes at the muscle level of a shift to more glycolytic metabolism by the mRNA sequencing data.  They relate much of this to changes in FOXO1 concentrations.  They make a strong case that FOXO1 is involved and show in a myotube preparation that alterations in FOXO1 lead to some of the changes note in the treated rats.  

The paper is well written as is and I have only a few minor comments that should be addressed

1.  The abstract should contain some selected most illuminating  quantitative data to give the reader some impression of the magnitude of changes.

2.  It is not entirely clear to me why serum starvation was used to force changes in the L6 myotube preparation and not hypoxia or Sugen hypoxia.  This relevance needs to be better justified.

3.   Why was diaphragm data not given in Figure 6? 

4.  In the discussion it is mentioned that therapeutic enhancement of FOXO1 might be utilized in some patients, but I don't follow the logic and it would be useful to describe how FOXO1 can be enhanced pharmacologically or genetically. 

Author Response

Please accept our responses. Thank you. 

Reviewer 2 Report

1.          The most important limitation of this study is; there is no direct evidence whether the numbers or ratio of type I/II fibers in soleus muscle were actually changed in SU/Hx animals. To visualize and count type I/II fibers directly, muscle sections should be subjected to NADH-TR staining or some other experiments. The authors need to indicate PAH-induced fiber conversion to clarify the purpose and claim of this study; because type I/II “markers” are expressed in one fiber. For example, the levels of type II markers are generally low in type I fibers, a little shift of absolute value can easily increase a fold-change. Alteration of marker expression in muscle tissue doesn’t guarantee the physiological or morphological conversion of muscle fibers.

2.          Western blottings in Figures 5C and 6D must be performed multiple times, quantitatively indicated, and statistically analyzed as Figure 2F.

3.          Supplementary Figure S1: Which part of the heart was immunostained? Left ventricle or other region? It should be described in the Results, Materials and Methods, and Figure Legends; because PAH involves right ventricular hypertrophy as the authors introduced in Line 31.

4.          Figure 4B: Numbers of fibers per field (4~6) are too little for statistical analysis. I wonder why fiber numbers were few even though high magnification images as in Figures 3B and 4B include dozens of fibers. For exact counting fiber numbers, lower magnification images containing hundreds of fibers should be utilized.

5.          Lines 147-150 and Figures 6C and 6D: Conflict results about FoxO1 expression between qPCR and Western blotting should be described and discussed in detail. Immunostaining of L6 myoblasts would clearly visualize (phospho-)FoxO1.

Author Response

Please accept our responses to your comments. Thank you.

Round 2

Reviewer 2 Report

The authors revised the manuscript according to the reviewer's comments.